

# The potential uses of tracer cycles for groundwater dating in
# heterogeneous aquifers
Julien Farlin[1], Piotr Małoszewski[2]
1. LIST Luxembourg Institute of Science and Technology, ERIN department, 41, rue du Brill, L-4422 Belvaux,
Luxembourg.
2. AGH University of Science and Technology Cracow, department of Hydrogeology and Engineering Geology, Al.
Mickiewicza 30, 30-059 Cracow, Poland
*Correspondence to*: Julien Farlin (julien.farlin@list.lu)
**Abstract**
The use of the annual cycles of stable isotopes to estimate the parameters of transit time distribution functions has been
recently criticised by Kirchner (2016). The author shows that the mean residence time of heterogeneous catchments
calculated from the damping of the amplitude of the input signal are very often over-estimates, sometimes by large factors.
We show here that the overestimation depends on the relative time scales of the cycle's frequency and the mean transit time
and that tracer cycles can still be used, at least for groundwater systems sustained by baseflow.
Firstly it appears that an exponential model is a good approximation for the transit time distribution of a heterogeneous
groundwatershed if the sub groundwatersheds' transit time distributions are themselves exponential and their mean transit
times are in the same range or slightly higher than the period of the tracer cycle.
Secondly, we suggest that tracer cycles can still be used as secondary data to test whether the degree of heterogeneity of the
subsurface is small enough to warrant approximating it by a homogeneous medium.
Lastly, we develop a model predicting the amplitude of groundwater temperature from the annual air temperature cycle, and
show that even though temperature is not a conservative tracer, it can be useful for groundwater dating. The potential use of
the temperature cycle is illustrated in the case-study of a sandstone aquifer drained by contact springs.
**1. Introduction**
Of the different methods that were developed to estimate groundwater transit time from environmental isotopes, fitting a
model to the damping of the amplitude of the atmospheric input (Maloszewski et al., 1983) is probably the most simple. It
only requires short times series of observations at the outlet (half a cycle at worst, i.e. half a year for the annual signal of
stable isotopes), and the fitting of a sine function. Until now, its main recognized drawback was the limited window of mean
transit time (MTT) accessible, as the function describing the damping of the amplitude with increasing MTT reaches the
quantification error of stable isotope measurements (about 1‰ for deuterium and 0.1‰ for [18]O) after only a few years.
The use of tracer cycles to estimate the parameters of the transit time distribution (TTD) function has been recently
reexamined by Kirchner (2016) for heterogeneous catchments. Using a simple model combining the transit time distribution
(TTD) of two theoretical sub-catchments characterised by a MTT diverging by at least a factor of two and calculating the
MTT from the resulting amplitude of the output tracer signal, he showed that the strong non-linearity of the relationship
between amplitude and MTT was liable to cause a systematic underestimation of the "true" MTT. These results contrast
with those of Luther and Haitjema (1998) who after conducting a series of numerical experiments using groundwater flow
models and spatially variable hydraulic conductivities, recharge and porosities, concluded that the exponential function





"provides a good approximation to the residence time distributions for watersheds with heterogeneitiy that is not significant
and distinct".
Since all catchments are to some degree heterogeneous, Kirchner's result seem to spell the end of the method altogether for
dating purposes. In this contribution we wish to build upon his excellent work and show that there still is room for
differentiation, at least for relatively slowly-responsive groundwater systems. To that effect, we adopt Kirchner's
methodology, which we slightly modify to shift the whole perspective from estimating the MTT to a more general
assessment of TTD estimation (in the vein of the paper by Luther and Haitjema).
This is done in order to study the effect of heterogeneity (in the sense of Kirchner's toy model) on the *shape* of the TTD, and
consequently on the robustness of the estimation of the MTT from tracer cycles. To this aim, we fit an exponential model
directly to the tracer output calculated from the transit time distribution (TTD) obtained from the sum of the sub-catchments'
TTDs in order to compare the (synthetic) real TTD with its exponential approximation. This comparison is done for
different degrees of heterogeneities and MTTs of different magnitudes relative to the period of the cycle (one year).
This analysis then leads to our suggestion of using tracer cycles as secondary data with which to test the "homogeneity
hypothesis"
We also introduce temperature as an alternative tracer for groundwater dating, and show that although it is not conservative,
its large seasonal amplitude and smooth annual cycle advantageously compare to stable isotopes and allow an extension of
the dating window to a mean residence time of about ten years in favourable cases. We illustrate the second and third points
with a case-study of a sandstone aquifer drained by contact springs.
**2. Modification of Kirchner's thought experiment**
Three remarks are of the order to begin with. Firstly, we assume that the tracer mean transit time is in first approximation
equal to the water mean transit time. Secondly, the mean transit time in itself is of limited interest to us, although useful
estimates can be gained from it, such as the total volume of water in storage or the average saturated porosity (Maloszewski
et al., 2002). But as a "characteristic measure" of a catchment's reaction time, it suffers from the same drawback as the
average value of any set of measurements in that it does not convey any information about the distribution of the transit time
around that mean value, which depends on the chosen TTD model. In our view, the mean transit time is simply a fitting
parameter of the TTD, and as such, much less appropriate than the *shape* of the TTD to study the effect of "heterogeneity"
on a catchment's response time. Thirdly, we restrict our analysis to groundwater systems in which the discharge in the outlet
is sustained exclusively by baseflow.
In order to study the effect of subcatchments' heterogeneities on the shape of a groundwatershed's TTD, we modify
Kirchner's methodology as follows. Firstly, we increase number of subcatchments to study a possible convergence of the
total TTD towards a stable distribution. Secondly, we do not enforce a factor 2 of difference between the sub-catchments'
MTTs (which Kirchner presents without further arguments as "truly heterogeneous catchments"). Instead, we investigate
the effect of the magnitude of subcatchments' MTTs *relative* to the period of the tracer cycle of one year.
The methodology is as follows:
1.  The number "n" of subcatchments is set
2.  An interval I for the subcatchments' MTTs is defined ($I \in \mathbb{R}_{>0}$).
3.  The MTT for each subcatchment is defined by drawing a random number from the interval I assuming a uniform

74          probability distribution.





4.  the TTDs of the subcatchments are added up and scaled to unity, yielding the TTD of the entire groundwatershed (TTD_composite)

5.  The input tracer cycle is convolved with the TTD_composite obtained, yielding an output signal characterized by a given amplitude damping.

6.  A theoretical exponential function is fitted to the output signal obtained from the TTD_composite, yielding a TTD_theory. This step simulates the real situation in tracer studies where the parameter of the TTD_theory is estimated inversely from the input and output signal.

The procedure is repeated for different subcatchments sizes n and for the different intervals I.

The function chosen for the TTD_theory is the exponential model, which was shown by Luther and Haitjema (1998) and Etcheverry (2001) to describe exactly the distribution of transit times in a homogeneous semi-confined aquifer for which the product nH/R (porosity times aquifer thickness divided by annual recharge) is piecewise constant. Another important result of Luther and Haitjema's numerical simulations is that mild heterogeneous hydraulic conductivities fields or unconfined aquifer with gently sloping groundwater tables lead to TTDs nearly indistinguishable from an exponential function as well. Following previous groundwater dating studies performed for the aquifer presented in the case study, a piston-flow component is added to the transfer function to simulate the transit time through the unsaturated zone (Farlin et al., 2013b).

## 3. Groundwater dating using the annual temperature cycles

To our knowledge, all publications using tracer cycles to estimate the parameters of a TTD are based on the seasonal signal of stable isotopes. One of the problems with that signal is its lack of smoothness. Depending on the year and the location, the isotopic seasonal variations in rainfall in some cases hardly qualify as sinusoidal. On the other opposite, air temperature and consequently soil temperature follow much more closely the seasonal pattern of the near-surface air temperature. In case of the soil temperature, soil thermal inertia imparts both a lag and a damping of the amplitude of the air temperature with depth which can be modeled by a sinusoidal function (Hillel, 1998)

$$T(z,t) = T_a + A_0 e^{-z/d} \sin\left[\frac{2\pi(t-t_0)}{365} - \frac{z}{d} - \frac{\pi}{2}\right] \qquad (1)$$

Where $T(z,t)$=soil temperature [°C] at depth z [m] and time t [d], $T_a$=average annual soil temperature [°C], $A_0$=annual amplitude of the surface soil temperature [°C], d=damping depth [m] and $t_0$=time lag from January 1 to the occurrence of the minimum temperature in a year [d]

The damping depth is given by

$$d = \sqrt{\frac{2D}{\varpi}} \qquad (2)$$

With D=thermal diffusivity of the soil [m$^2$.s$^{-1}$] and ω=2π/365 [d$^{-1}$]

Assuming sufficient time for infiltrating water to come to thermal equilibrium with the soil, equation 1 can be used to predict soil water temperature as a function of depth and of the thermal properties of the soil.

After leaving the soil compartment, the additional damping of the temperature signal that occurs during transport in the aquifer can be modeled by the following convolution integral (Maloszewski and Zuber, 1982)

$$T_{out}(t) = \int_0^t T_{in}(t) g(t-\tau) \exp(-\lambda\tau) d\tau \qquad (3)$$



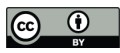

Where $T_{out}(t)$=water temperature in the outlet at time t [°C], $T_{in}(t)$=water temperature at the groundwater table at time t [°C],
g=TTD$_{theory}$, and λ=loss constant [d$^{-1}$]. λ usually simulates decay for radioactive tracers such as tritium or degradation for
non-conservative tracers. For temperature, we assume long-term equilibrium with the surrounding rock, and set λ=0.
The exponential piston-flow TTD is given by
$$g(t) = \frac{\eta}{\upsilon}\exp(-\frac{\eta t}{\upsilon} + \eta - 1) \text{ for } t \geq \upsilon\left(1 - \eta^{-1}\right)$$    (4)
$g(t) = 0$ otherwise
With υ=MTT [d] and η=total volume divided by the exponential flow volume [-]
Thus, the temperature signal in water at the outlet of the groundwatershed can be modeled from the annual cycle of air
temperature transformed in series first by its transfer first through the soil (equation 1) and then through the aquifer
(equation 3). Equation 3 can also be used with another tracer, in which case $T_{out}(t)$ is the tracer concentration in the outlet. In
any case, $T_{out}(t)$ and $T_{in}(t)$ are known, and υ and η are the two free parameters that must be estimated by curve fitting. The
complete soil-aquifer model developed here consists of 4 free parameters: the soil depth z, the mean soil thermal diffusivity
D, the MTT υ and the piston-flow parameter η.
**4. Field sites**
The fractured-rock aquifer known as the Luxembourg Sandstone is the main groundwater resource of the country of
Luxembourg. The aquifer, unconfined in its northern part, is drained by numerous contact springs emerging where its
impervious basis intersects the land surface. More details about the aquifer can be found in Farlin et al. (2013a).
Water temperature and stable isotopes were measured weekly in 11 contact springs draining the Luxembourg Sandstone
aquifer between May 2013 and July 2014. Additionally, between 1 and 5 tritium measurements were also available
depending on the spring, covering the period 2008-2014. The atmospheric input functions for tritium and $^{18}$O was created by
extending backwards the monthly measurements of the station Trier available from 1978 to the present with the monthly
measurements of the station Vienna using linear regressions. The input time series were further weighted by the ratio of
summer to winter infiltration coefficient (Grabczak et al., 1984) using the stable isotope measurements.
The TTD of each spring was parameterized jointly using tritium and $^{18}$O measurements. Only parameters that lead to output
signals within the analytical error of both tracers were retained, yielding a range of estimated MTTs for each spring. The
stable isotope signal could not be used for sine fitting as it was either flat, or showed an interannual trend depending on the
spring.
The mean annual air temperature in the region is 10 °C with an annual amplitude also equal to 10 °C. The minimum air
temperature is usually reached at the end of January, so $t_0$ is set to 30 days. Soil thermal diffusivity depends on mineralogy,
water content and bulk density. For a bulk density of 1.5 Mg.m$^{-3}$, measured D vary between to 0.2*10$^{-6}$ and 10$^{-5}$ m$^2$.s$^{-1}$
(deVries, 1963;Farouki, 1986). Furthermore, the depth of the soil also influences the damping of the temperature signal.
Soil depth in the catchments is variable and can reach 2 meters (Farlin et al., 2013a). Since the mean value of D and z
cannot be estimated separately for each groundwatershed, calculations were made using minimum and maximum known
values, yielding bounding curves in the plot of temperature amplitude versus MTT.



## 5. Results

Three different intervals were used for the TTD estimations. The interval adopted by Kirchner ($I_3$ : 0.1-20 years), as well as two sub-intervals ($I_1$ : 0.1-1 year and $I_2$: 5-20 years). Figure 1 shows the result of the comparison of the $TTD_{composite}$ with the $TTD_{theory}$ for five different realisations of the $TTD_{composite}$ for the three different MTT ranges. Results are insensitive to the number of subcatchments and are shown exemplarily for n=10. The realisations shown are a small subset of all the simulations performed chosen to illustrate the different TTD shapes obtained.

Both for short MTTs ($I_1$) and the entire MTT range adopted by Kirchner ($I_3$), the obtained TTDs display a significant departure from an exponential function. All simulated TTDs are more strongly curved than the exponential distribution. For MTTs shorter than the period of the annual cycle ($I_1$), the younger fraction dominates in the output signal which causes the exponential function to follow the slope of that fraction. When MTTs are drawn from the entire range of 30 days to 20 years ($I_3$), the slope of the fitted exponential function often lies between the two segments of the $TTD_{composite}$ in order to balance out the respective influence of the "young" and "old" fractions. As a consequence, the young water contribution is overestimated, while the opposite holds for the old water contribution. The situation is different for the medium MTT range ($I_2$). In that case, the entire $TTD_{theory}$ obtained inversely from fitting follows closely the $TTD_{composite}$ over the entire interval of transit times of 0 to 20 years.

A plot of the measured amplitude versus the calculated MTT together with the theoretical curves obtained for different parameterisations of the soil compartment is presented on Figure 2. Seven out of eleven springs fall within the interval spanned by the theoretical curves (eight out of eleven taking the error bars into account). For the last three springs, the measured amplitude of water temperature is higher than expected from the calculated MTTs. Two of these springs probably drain a more superficial aquifer than the others, and thus may display a TTD more curved than the exponential model of the type shown for the interval $I_3$ on Figure 1. Figure 2 also illustrates the fact that for a shallow soil and an amplitude of the input temperature signal of 10°C, MTTs of up to 20 years can be estimated before the measurement error of ±0.1°C is reached.

## 6. Discussion

Our re-exploration of Kirchner's model has shown that (i) as long as the range of subcatchments' MTTs is in the same order of magnitude as the period of the cycle, and supposing all TTDs are exponential, the total TTD can still be approximated by an exponential distribution and consequently that (ii) although using tracer cycles to estimate the parameter of a TTD must be used with caution and acknowledging the heterogeneity problem, the method still appears sufficiently robust for particular applications such as groundwater studies as long as the quickflow contribution to discharge is negligible compared to baseflow.

The first point confirms the findings of Luther and Haitjema concerning heterogeneities that are "not significant and distinct" using Kirchner's simple toy model. As Kirchner puts it, "it will be generally difficult or impossible to characterize the system's heterogeneity". We can only agree, and wish to add that this shortcoming becomes much less glaring if a catchment's heterogeneity is simulated using a groundwater flow model as was done by Luther and Haitjema (1998), since the variables' physical parameters (porosity, hydraulic conductivity and recharge rate) can be compared to real-world measurements. This compares favourably to the type of guesswork necessary for a model based on combining subcatchments TTDs, where the "likely range of variation in subcatchments MTTs"(Kirchner, 2016) is difficult to estimate, and which Kirchner supposes to go from "fractions of a year to perhaps several years" without substantiating his claim.



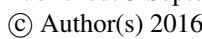

The second point introduces a degree of differentiation to the results obtained by Kirchner. Our analysis shows that the
decisive point is not so much the difference between sub-catchments MTTs, but rather its magnitude relative to the time
scale of the cycle used for dating. Even when individual MTTs differ by a factor of up to 4, as long as they are drawn from
an interval slightly higher than the period of a cycle (in most cases one year), the total TTD follows closely an exponential
function if the individual TTDs are themselves exponential. This is not surprising considering the strong smoothing effect
exercised by the TTD on the input function.
We also present a groundwater dating model based on the annual temperature cycle by combining a function describing soil
temperature as a function of soil depth and the traditional convolution integral of Maloszewski and Zuber (1982).
Temperature is not a conservative tracer, and as such suffers from a number of drawbacks. Firstly, Eq. (1) is strictly valid
using soil surface temperature, and not air temperature as we have done. Wu and Nofziger (1999) however have shown that
this approximation leads to a systematic underestimation of soil temperature for bare soils, which can be corrected simply
by adding 2 degrees to the prediction. Since our analysis is based solely on the amplitude, and not on absolute temperature,
such a correction is unnecessary. Secondly, Eq. (1) predicts soil temperature, and not water temperature. Thus we had to
assume thermal equilibrium between the soil and the infiltrating water. This approximation of course should not hold if fast
preferential flow is the main infiltration mechanism. However, since we calculate water temperature at the soil-bedrock
interface, and we expect the density of preferential flow pathways to decrease with depth, the model assumption appears
reasonable in the absence of actual field measurements of water temperature at such depth. Thirdly, the natural geothermal
gradient imparts an increase of subsurface temperature of 1°C per 20 to 40 m (Anderson, 2005). As for the difference
between air temperature and soil surface temperature, an increasing heat content is not problematic as long as it is constant
in time, since only the amplitude of the temperature signal is used in the analysis.
On the other hand, the advantages of the temperature cycle should not be overlooked. Temperature measurements in
groundwater or spring water are often readily available, measurement is cheap and can be done automatically using field
probes, and one cycle measured at the outlet is enough to estimate the MTT parameter of the TTD. Furthermore, both input
and output signals are much smoother than the often noisy stable isotope cycles.
The case study we present constitutes a first test of our theoretical analysis. The groundwatersheds of the springs sampled
appears in eight out of eleven cases to be sufficiently homogeneous to allow approximating their TTDs with an exponential
function. Furthermore, the case study illustrates how tracer cycles can be used to verify whether the aquifer is locally
sufficiently homogeneous for such an approximation. The fact that three springs fall outside of the envelope of Figure 2
could indicate that the underlying exponential TTD fails to represent the true TTD of the three groundwatersheds. As we
have seen, this situation can be observed in the presence of strong local heterogeneities leading to a TTD more strongly
curved than an exponential curve. In such a case the estimated MTT is meaningless, since it parameterises a function that
underestimates the old water component and often overestimates the young water component as well.
In conclusion, we recognize two uses for tracer cycles:
• In subsurface systems that react relatively slowly to precipitation events, the cycle can be used for dating purposes.
Water temperature in particular can be used in that way when a more complex approach based on costlier tracers
cannot be adopted.
• In dating studies making use of tracers such as tritium or CFC measurements, the additional MTT estimates
obtained from the amplitude damping can be compared to those calculated from the environmental tracer. A
disagreement could indicate that the chosen TTD is inappropriate, possibly because of strong and distinct local





heterogeneities within the groundwatershed.

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





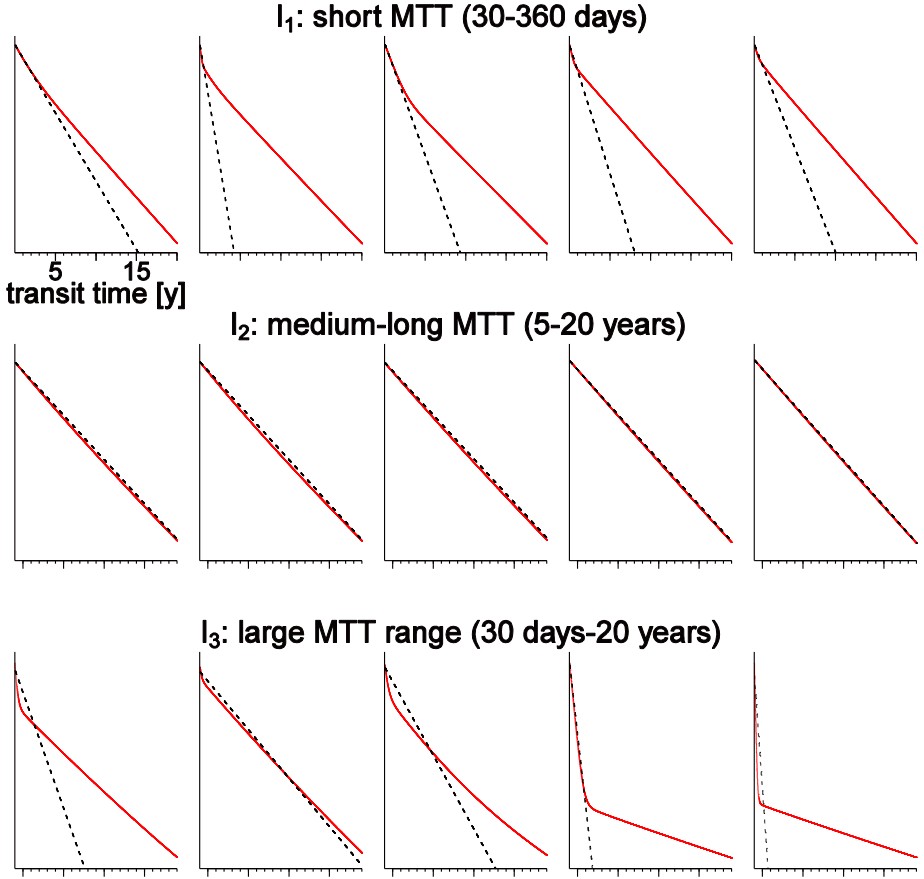

**Figure 1: Comparison of the TTD$_{theory}$ (black dotted line) and TTD$_{composite}$ (plain red line) for different MTT ranges (exponential y-axis)**





**Figure 2: Amplitude damping of the temperature signal as function of the MTT and for different values of D and z, and observed relationships for the sampled springs. The horizontal error bars mark the range of MTTs estimated using tritium and [18]O measurements, the vertical error bars correspond to the instrumental error of the temperature probe (0.1 °C).**

**Author contribution**
J. Farlin performed the field work and the calculations and wrote the manuscript. P. Małoszewski added elements to the
draft.
**Competing interest**
The authors declare that they have no conflict of interest.