# Peer review of "The potential uses of tracer cycles for groundwater dating in # heterogeneous aquifers"

_Hydrology and Earth System Sciences, 2016_

## Referee Comment (RC1) · Anonymous Referee #1 · 28 Sep 2016

Reviewer's comments for HESS-2016-393

Farlin and Maloszewski, 2016, The potential uses of tracer cycles for groundwater dating in heterogeneous aquifer, Hydrol. Earth Syst. Sci. Discuss.

Tracer test techniques are widely used for groundwater dating applications. The authors provided three points of arguments and suggestions on the hydrogeology tracer tests, based on some previous studies. Generally speaking, most statements in this manuscript have been discussed in many previous literatures, in other words, not novel. The manuscript failed to explain the statements clearly with enough information. This paper is still far away from the quality to be published, especially the figure. I would suggest a major revision, but recommend the editor to take an additional look since the

required revision will be large. Please see my detail comments as follow.

1) Title: The authors use so many words and pages in the manuscript to discuss the groundwater temperature tracer applications, however, not mention it on the title. The third statement of groundwater temperature model is obviously the most important part in this paper. It is necessary and worthy to stress on the title.

2) Ln 15-16: Why exponential model? Can you provide any mathematic and/or statistic analysis on different fitting model? I would also suggest to take a look/cite more study results in previous literatures.

3) Ln 18-19: The authors discussed the degree of heterogeneity is small enough or not for tracer cycles. How to define the degree of heterogeneity? Is there any quantity definition?

4) Ln 20-22: Groundwater temperature tracer has been widely used in so many previous studies. It's not a new technique at all. I'm not quite sure about the authors' major contribution and breakthrough in modeling development of temperature tracer application, which should be highlighted in the abstract.

5) Ln 41: Groundwater system is not always "slowly-responsive". The authors probably need to be specified their study area as "sandstone aquifer".

6) Ln 66-80: Detail explanations are needed for the modification of Kirchner's methodology. For example, what are the benefits of increasing number of sub-catchments? Why do you use a random number as MTT? Is the result sensitive to the number?

7) Ln 123-125: A more detailed introduction of field site is necessary. You can't only cite previous paper without enough information.

8) Ln 130-131: What are the time spans of summer/winter periods?

9) Ln 146-147: The authors claimed that results are insensitive to the number of sub-catchments. Why did you still need to divide sub-catchment?

10) Ln 170-171: How to determine the method is "sufficiently robust"? Any quantity? A statistic analysis will be important here.

11) Ln 184-185: Is it redundant to state the total TDD exhibits the same properties with individual TDDs? I think this is obvious. Some explanation is necessary if the authors believe there is any scientific contribution here.

12) Ln 196: The word "density" is misleading here and easy to confuse readers.

13) Ln 214-216: The first conclusion is obviously not new found.

14) Figure 1: The y axis and label missing. The MTT range in the third row (Kircher method) is actually between I1 and I2, so it's not right to call it large MTT range. This figure quality is not acceptable.

---

## Referee Comment (RC2) · Anonymous Referee #2 · 5 Dec 2016

This manuscript presents a discussion on the use of tracer cycles in analyzing transit time distribution for heterogeneous aquifers with the problems that was figured out by Kirchner (2016). The authors attempt to show different results of hypothetical examples that were simulated with the method similar to Kirchner (2016). Further, the authors suggest the use of temperature cycle as a tracer with significant seasonal cycles. This may be a contribution for the selection of a sufficient tracer with cycle signals. However, I am confused by the presentation with the diverging information and could not capture what are highlighted and concerned from the discussions.

Major Comments:

1.  The manuscript just seems to be a discussion rather than a full-pages research

article. The method and materials are not presented with enough contents to be followed, especially for the experimental studies and the case study of the Luxembourg Sandstone.

2. The first conclusion in the manuscript is that the exponential model is successful for the mixed signal when the mean transit times (MTT) of sub-basins are "in the same range or slightly higher than the period of the tracer cycle". This conclusion seems come from group I2 in Figure 1 but it is confused because MTTs in group-I2 are 5-20 years, exactly not "slightly higher" than the cycle (1-year). In addition, this could not be considered as a general feature because the experiments are hypothetical and special with a uniform probability distribution for the MTTs interval.

3. The second conclusion is "the tracer cycles can still be used as secondary data" to exam "the degree of heterogeneity". I don't think it is a conclusion but just a suggestion and seems to be not a substantial contribution for the topic.

4. The method of using groundwater temperature has not been clearly presented, especially for the case study. It is hard to understand why the performance of temperature is well in the studied sites. Figure 2 shows significant scatter distribution (the number of samples is also small) from which the performance is difficult to be evaluated.

Minor Comments:

1. Line 72, what is the mean of R>0?

2. Line 138, 0.2*10 ?

3. Miss titles of coordinates and labels in Fig. 1.

Reference:

Kirchner, J. W.: Aggregation in environmental systems-Part 1: Seasonal tracer cycles quantify young water fractions, but not mean transit times, in spatially heterogeneous catchments, Hydrology and Earth System Sciences, 20, 279-297, 2016.

---

## Author Comment (AC1) · 2 Jan 2017

Dear reviewers, dear Editor,

Following the reviewers' comments and the recent publication of the manuscript by Stewart et al. on the aggregation error of tritium-based transit times in HESS discussions, we will modify the text to emphasize the disagreement between the findings of Kirchner and those of Luther and Haitjema and make this the core of the article. We will also change our conclusion slightly, since we realised that mean transit times shorter than the period of the tracer cycle cause a deviation from the exponential model (see answer 2 to reviewer #2). We will also to put more emphasis on the one major difference between our approach and Kirchner's: we concentrate on the shape of the transit

time distribution while he studied mean transit times only. We feel this important point was missed by both reviewers, so will try to explain it better. We argue that focussing on the MTT instead of understanding the underlying reasons by looking at the shape of the TTD is a mistake and a source of confusion and should be avoided. The manuscript consists of two parts, first a theoretical reexploration of Kirchner's model, then the introduction of a method with which to use a particular tracer cycle, temperature. Both parts then come together to illustrate the two uses that can be made of that particular cycle for groundwater dating (as standalone if other tracers are not available, and as additional data with which to test the MTT estimates obtained from an environmental tracer (tritium for instance) and possibly calibrate a compound lumped parameter model (this is new and will be explained below in the answer 3 to reviewer #2). We regret that the second part, namely the temperature model, has hardly been reviewed by the two referees. Here are our answers to their specific criticisms.

Reviewer #1

Reviewer #1 reproaches the lack of novelty of the manuscript, and we feel that either we have failed to explain properly and clearly the aim of the study, or the reviewer has failed to grasp it (maybe he has read the text too quickly, as some of his specific remarks suggest). We do not claim to be breaking new ground here, but to (i) try and explain why the findings of Kirchner seem to contradict those of Luther and Haitjema, the first showing that "heterogeneities" generally cause a deviation of the transit time distribution sufficient to affect parameter estimation, the latter showing that in many cases that deviation is small and hence negligible for practical applications, and (ii) to introduce the coupling of the common lumped parameter model with a soil model describing the damping of the amplitude of recharge water with depth. Neither point has to our knowledge been discussed before in the large literature on water dating. We will put more clearly the emphasis on the contradiction between Kirchner's and Luther and Haitjema's results right from the introduction, which should improve general readability by providing a clear scientific question.

1) This is a good point. We will modify the title to emphasize the importance of the temperature cycles used for groundwater dating. By the way, we regret that neither reviewer has really commented specifically on the temperature model we propose.

2) The choice of the exponential model will be substantiated more thoroughly than in the first manuscript. We will make clear that we have limited ourselves to the exponential TTD because previous papers have shown it to be an exact or at least robust approximation for many different types of aquifers and because this is the important special case that Luther and Haitjema studied. Additionally, it is independent of aquifer shape, as Haitjema has shown in his 1995 paper in Journal of Hydrology. We will add this citation to the manuscript. We will also emphasize that the study is strictly considering groundwater systems (where the exponential TTD is a reasonable assumption), and not whole catchments (where evidence rather points to a gamma TTD).

3) Since what we mean by "degree of heterogeneity" is explained in the article, we feel its use in the abstract conveys the idea succinctly. An indirect quantification of that degree of heterogeneities is found in the results and discussion sections, where we point out to the fact that the TTD does not deviate significantly from an exponential function as long as the MTTs of the subcatchments are not shorter than a the period of the tracer cycle (one year).

4) Groundwater temperature is indeed a very common tracer in groundwater studies, but NOT in the way we use it in this study. We do not claim that the tracer is new (and we cite the review paper by Anderson on temperature as groundwater tracer), but present a technique with which to calculate the output damping of the temperature signal of recharge water due to both the soil and the aquifer compartment, or use both input and output signals inversely to calibrate the TTD of the aquifer. If this particular coupling has already been published, we would be very grateful to the reviewer to give us the reference. We will modify the wording of the abstract to make the use of the temperature model clearer to the reader.

5) This reference to "slowly-responsive" systems was clumsy and will be deleted in the general overhaul of the paragraph, and replaced by a discussion of the validity of the exponential model for aquifers systems.

6) We do not understand how our modification of Kirchner's method is unclear. We stated explicitly that the number of subcatchments is increased to study whether this leads to an "averaging out" of the subcatchments' TTDs (it turned out that is does not, but this was not intuitive to us). The use of random number is NOT a modification, but follows Kirchner's method (except that he used a log scale and we did not). And we do show that the overestimation depends on the numbers, or rather on the intervals from which the MTTs are drawn.

7) The description of the study site will be extended slightly to include information on soil type and recharge, and a map of the area added to the manuscript. We would have appreciated to know more precisely what kind of information the reviewer felt was missing.

8) The winter period runs from October to March of the following year, and the summer period from April to September.

9) As we wrote in answer to question 6), it was not clear to us whether increasing the number of subcatchments would lead to a smoothing out of the total TTD. With hindsight it makes sense that it does not, but at that time, it did not, and we supposed some readers might be curious to know, or tempted to try it out.

10) We meant with "robust" that for the simulations we performed, the total TTD deviated negligibly from an exponential function as long as the individual MTTs were within the interval 5-20 years , and consequently that the TTD calibrated using the output tracer signal was very close to the "true" synthetic one. We did not use the concept of robustness in its strict statistical sense (and even in statistics, robustness is often NOT measurable, but relies on comparison between methods).

11) Stating that the total TTD is approximately exponentially distributed when the sub-catchments are exponential is not redundant at all, since Kichner in his paper makes the exact opposite point and shows for instance that the sum of two exponentials is not an exponential but a hyperexponential. What we mean to say is that even though the sum strictly speaking follows another shape, it deviates so little from an exponential than in practice it does not matter. A sentence explaining this will be added to the discussion.

12) We will change "density" for "importance".

13) We will rephrase the first conclusion, hoping it will be clearer that this is the MAIN results of the re-exploration of Kirchner's model. His conclusion was that tracer cycles cannot be used in heterogeneous catchments to calibrate a lumped parameter model because of the overestimation of the MTT due to the nonlinear relationship between MTT and tracer amplitude. We conclude the OPPOSITE and say that Kirchner's conclusions were too broad and need to be nuanced.

14) We would have been grateful to know more precisely in what way the quality of the figure is not acceptable so as to amend it, but will try to guess what the reviewer disliked. The labels were not missing, but omitted on all figures but the first since the scaling is the same on both axes (which is also why the numbering on the x axis is present only on one graph). We will movedthe label to the bottom left figure and mention it in the caption. We will also add a y axis, realising its meaning was not obvious. What does the reviewer mean with "in between" ? I1 ranges from 30 to 360 days, I2 from 1 to 20 years and I3 from 30 days to 20 years. So I3 spans the entire range, while I1 only covers shorter MTTs and I2 longer ones. This is what we mean by "large", and do not see what is wrong with the term in that situation.

Reviewer #2

Major comments:

1) We felt a short manuscript was sufficient to convey our experiment. We will extend the discussion and the description of the study site slightly, but wish to keep the text compact and to the point.

2) This is a major mistake in the first manuscript. We have redone the calculations using an interval I2 now spanning the range 1 year to 20 years, thus with a lower bound equal to the period of the tracer cycle. This is important, since we show that deviations from the exponential model are caused by shorter MTTs (below one year). We do not understand the reviewer's second criticism. The experiments are "hypothetical" in the sense that we have used a simple conceptual model to simulate heterogeneous catchments, but the method is in that respect exactly similar to Kirchner's (which we do criticise in the discussion). The numerical simulations are synthetic, but meant to be generic enough to shed light on the effect of subcatchments' mixing on the TTD (our paper) or on the estimation of the total MTT (Kirchner's paper) in a very general sense. And what makes sense more than a uniform probability distribution for the random selection of MTTs, since no MTT should be weighted more than the others ?

3) We will reformulate what was indeed a common sense suggestion by introducing the idea that compound lumped parameter models (a double exponential) could be used when one suspects distinct heterogeneities to cause a deviation from a simple exponential model (taking into account the hydrogeological setting), as suggested by Stewart et al. in their manuscript on aggregation error for tritium-based MTTs recently submitted in HESS discussions. We will suggest that the components of a double exponential model could be calibrated separately using the tracer cycle for the younger fraction and tritium for the older, since we have shown on figure 1 that the fit obtained from the cycle follows closely the younger fraction. It is not a revolutionary idea, but well worth repeating, judging from the confusion still surrounding model choices.

4) Does the reviewer mean that the theory is not clear, or that our results seem unconvincing ? We feel the paragraph presenting the temperature model (paragraph 3) is systematic and to the point. As for the case study, we used the data at our disposal, and will welcome additional testing from others on richer datasets. We will add a sentence in the discussion to emphasize the need for more testing of the temperature model.

Minor comments:

1) This is standard mathematical notation for strictly positive real numbers.

2) We will change 0.2*10-6 to 0.2x10-6

3) This was also criticised by reviewer #1 and will be change accordingly.

Best regards.

Julien Farlin and Piotr Małoszewski